# Glucose Induces Resistance to Polymyxins in High-Alcohol-Producing *Klebsiella pneumoniae* via Increasing Capsular Polysaccharide and Maintaining Intracellular ATP

Zheng Fan,[a] Tongtong Fu,[a] Hongbo Liu,[b] Zhoufei Li,[a] Bing Du,[c] Xiaohu Cui,[a] Rui Zhang,[a,d] Yanling Feng,[a] Hanqing Zhao,[a] Guanhua Xue,[a] Jinghua Cui,[a] Chao Yan,[a] Lin Gan,[a] Junxia Feng,[a] Ziying Xu,[a] Zihui Yu,[a] Ziyan Tian,[a] Zanbo Ding,[a] Jinfeng Chen,[a] Yujie Chen,[a] Jing Yuan[a]

aDepartment of Bacteriology, Capital Institute of Pediatrics, Beijing, China
bSchool of Pharmaceutical Sciences, Jilin University, Changchun, China
cUniversity of Edinburgh, Edinburgh, United Kingdom
dGraduate School, Peking Union Medical College, Beijing, China

Zheng Fan and Tongtong Fu contributed equally to this article. Author order was determined on the basis of seniority.

**ABSTRACT** High-alcohol-producing *K. pneumoniae* (HiAlc *Kpn*) causes nonalcoholic fatty liver disease (NAFLD) by producing excess endogenous alcohol in the gut of patients with NAFLD, using glucose as the main carbon source. The role of glucose in the response of HiAlc *Kpn* to environmental stresses such as antibiotics remains unclear. In this study, we found that glucose could enhance the resistance of HiAlc *Kpn* to polymyxins. First, glucose inhibited the expression of *crp* in HiAlc *Kpn* and promoted the increase of capsular polysaccharide (CPS), which promoted the drug resistance of HiAlc *Kpn*. Second, glucose maintained high ATP levels in HiAlc *Kpn* cells under the pressure of polymyxins, enhancing the resistance of the cells to the killing effect of antibiotics. Notably, the inhibition of CPS formation and the decrease of intracellular ATP levels could both effectively reverse glucose-induced polymyxins resistance. Our work demonstrated the mechanism by which glucose induces polymyxins resistance in HiAlc *Kpn*, thereby laying the foundation for developing effective treatments for NAFLD caused by HiAlc *Kpn*.

**IMPORTANCE** HiAlc *Kpn* can use glucose to produce excess endogenous alcohol for promoting the development of NAFLD. Polymyxins are the last line of antibiotics and are commonly used to treat infections caused by carbapenem-resistant *K. pneumoniae*. In this study, we found that glucose increased bacterial resistance to polymyxins via increasing CPS and maintaining intracellular ATP; this increases the risk of failure to treat NAFLD caused by multidrug-resistant HiAlc *Kpn* infection. Further research revealed the important roles of glucose and the global regulator, CRP, in bacterial resistance and found that inhibiting CPS formation and decreasing intracellular ATP levels could effectively reverse glucose-induced polymyxins resistance. Our work reveals that glucose and the regulatory factor CRP can affect the resistance of bacteria to polymyxins, laying a foundation for the treatment of infections caused by multidrug-resistant bacteria.

**KEYWORDS** HiAlc *Kpn*, resistance, polymyxins, *crp*, ATP, HiAlc Kpn

Address correspondence to Jing Yuan, yuanjing6216@163.com.

The authors declare no conflict of interest.

*Klebsiella pneumoniae* is an opportunistic bacterial pathogen that causes various infections, including urinary tract infections, hospital-acquired pneumonia (HAP), bacteremia and liver abscesses (1–3). Emergence of multidrug-resistant *K. pneumoniae* strains, such as carbapenem-resistant *K. pneumoniae* (CRKP), greatly increases the difficulty of clinical treatment (4). Polymyxins (i.e., polymyxin B [POL] and colistin [COL]) are

often used as the last line of antibiotic defense to treat infections caused by CRKP (5). Polymyxins act by electrostatically binding to lipid A, a component of lipopolysaccharide (LPS), and this binding helps polymyxins insert their hydrophobic domains into the bacterial fatty acyl chain of LPS and the inner membrane leaflet, thereby destroying the normal function of the bacterial membrane and causing bacterial death through the leakage of cytoplasmatic contents (6, 7). Furthermore, polymyxins can inhibit bacterial respiration by inhibiting the type II NADH-quinone oxidoreductase, depleting ATP and eventually causing bacterial death (8–10).

The bacterium has a variety of resistance mechanisms against polymyxins, including the loss of LPS (11), efflux pumps (AcrAB and KpnEF) (12, 13), overexpression of the outer membrane protein (OprH) (14, 15), and various LPS modifications, such as modifications of lipid A with 4-amino-4-deoxy-L-arabinose (L-Ara4N) or phosphoethanolamine (PEtN) (16, 17). Capsular polysaccharide (CPS) is an important virulence factor and has been shown to contribute to polymyxins resistance in *K. pneumoniae*. Purified CPSs from *K. pneumoniae* can increase resistance to polymyxins in unencapsulated *K. pneumoniae*, and CPS mutant strains are more sensitive to polymyxins than wild-type strains (18, 19). This happens because CPS and polymyxins attract each other by electrostatic interaction, reducing the amount of polymyxins that can come into contact with the lipid A component of the LPS on bacterial outer membrane (18).

Microbiome-derived alcohol has been shown to be closely associated with the development of NAFLD (20). Previously, we found that high-alcohol-producing *K. pneumoniae* (HiAlc *Kpn*) caused NAFLD (1). After consuming an alcohol-free, high-carbohydrate diet, the blood alcohol concentration of a patient with NAFLD can reach 400 mg/dL, which results from the excess endogenous alcohol produced by HiAlc *Kpn* in the intestine using carbon sources such as glucose. In our previous study involving a Chinese cohort, it was revealed that HiAlc *Kpn* is present in the intestines of 60% of patients with NAFLD (1). Antibiotic treatment is effective in alleviating symptoms in patients; however, the presence of glucose could alter bacterial resistance to antibiotics. In *Escherichia coli*, glucose can increase the uptake of aminoglycosides by promoting proton-motive force, which further enhances the killing of persisters (21). Exogenous glucose can restore susceptibility to kanamycin in both Gram-positive and Gram-negative bacteria by promoting the tricarboxylic acid (TCA) cycle and stimulating the uptake of antibiotics (22). Polymyxins can kill Gram-positive bacteria by consuming intracellular ATP, a process that can be interrupted by high glucose concentrations via maintaining bacterial ATP levels (10). However, the effect of glucose on the antibiotic resistance of HiAlc *Kpn* is unknown.

In this study, we found that glucose was able to induce polymyxins resistance in HiAlc *Kpn*. Further studies showed that glucose could increase the CPS content by decreasing the expression of *crp*. In addition, *crp* mutant exhibited substantially enhanced resistance to polymyxins by promoting CPS production. Moreover, glucose maintained high ATP levels in HiAlc *Kpn* treated with polymyxins, effectively increasing bacterial survival. Therefore, our results reveal the important roles of glucose in polymyxins resistance in *K. pneumoniae*.

## RESULTS

**Glucose induced resistance to polymyxins in HiAlc *Kpn*.** To test the role of glucose in the antibiotic resistance of HiAlc *Kpn*, we determined the MICs of various antibiotics against wild-type (WT) strain under 2% glucose. When treated by glucose, WT strain showed a higher-level resistance (MICs) to polymyxins, including POL and COL (Table 1). Using polymyxin B as a typical representative of polymyxins, consistent with the MIC test results, bacterial survival rate increased significantly in the presence of glucose (Fig. 1).

**CRP influenced bacterial resistance to polymyxins in HiAlc *Kpn*.** In *K. pneumoniae*, exogenous glucose could modulate the cell morphology and the CPS biosynthesis through the global regulator CRP (23, 24). However, it is unclear whether CRP could regulate polymyxins resistance in *K. pneumoniae*. To test the role of CRP in antibiotic resistance of HiAlc *Kpn*, the majority of *crp* coding region was deleted from the WT by allelic exchange to generate the Δ*crp* mutant. Deletion of *crp* resulted in slower

**TABLE 1** Bacterial susceptibilities to polymyxins (standard inoculum: $1 \times 10^5$ CFU/well)

| Antibiotic | MIC ($\mu$g /mL) | | |
| --- | --- | --- | --- |
| | HiAlc *kpn* (W14) | LowAlc *kpn* | *E. coli* ATCC 25922 (QC) |
| COL | 2 | 2 | 1 |
| COL (+2% Glc) | 4 | 4 | 2 |
| POL | 2 | 2 | 1 |
| POL (+2% Glc) | 4 | 8 | 2 |
| POL (+2.0 mM Na$_2$ATP) | 8 | 8 | 2 |

bacterial growth in both solid and liquid cultures, which was consistent with previous finding (Fig. 2A and B) (25). The MICs of POL and COL were increased 4-fold in the Δ*crp* mutant and complementation with *crp* restored the bacterial susceptibility (Table 2). Consistent with the MIC test results, in the presence of 8 $\mu$g/mL POL, deletion of *crp* increased the bacterial survival rate by approximately 1,000- to 10,000-fold, which was restored by complementation with *crp* (Fig. 2C). These results indicated that CRP could regulate polymyxins resistance in HiAlc *Kpn*.

The regulator CRP was known to affect capsule-associated phenotypes and regulate the *cps* expression (23, 25). To confirm that CRP could affect biosynthesis of CPS in HiAlc *Kpn*, we detected levels of the mucoviscosity and the amounts of capsular uronic acid in the WT and Δ*crp* mutant strain. The CPS content was significantly increased in Δ*crp*, and complementation with *crp* restored the CPS content (Fig. 2D and E). There are three typical promoters in the *cps* locus, located, respectively, on the upstream of *galF*, *wzi*, and *manC* (Fig. 2F). Compared to WT, the mRNA levels of CPS biosynthesis genes were significantly increased in the Δ*crp* (Fig. 2G). In addition, the high survival rate of the Δ*crp* mutant strains under POL treatment was significantly decreased following treatment with sodium salicylate and EGTA as they can inhibit the formation of CPS (Fig. 2H). Gene *manC*, which encodes mannose-1-phosphate guanylyltransferase, has been reported to affect the formation of CPS in *K. pneumoniae* (18). Deletion of the *manC* gene also significantly reduced the high survival rate of the Δ*crp* mutant strain when treated with POL (Fig. 2I).

**Downregulation of *crp* contributed to glucose-induced polymyxins resistance in HiAlc *Kpn*.** It has long been established that environmental glucose could inhibit the intracellular levels of cyclic AMP (cAMP) and the transcription of *crp* in bacteria (24, 26, 27). Hence, the above results indicated that glucose could inhibit the expression of *crp*, promote the production of CPS, and then induce the resistance of HiAlc *Kpn* to polymyxins. As expected, the exogenous glucose could inhibit the expression of *crp*, which was restored by addition of cAMP in HiAlc *Kpn* (Fig. 3A). Glucose increased the formation of CPS in HiAlc *Kpn*, adding the exogenous cAMP significantly reduced the CPS level (Fig. 3B and C). In addition, under high concentration of glucose, the exogenous cAMP significantly reversed

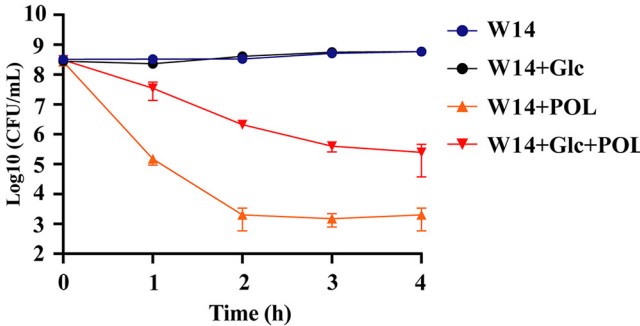

**FIG 1** Glucose induced resistance to polymyxins in HiAlc *Kpn*. The WT strain was grown to an OD$_{600}$ of 0.8 to 1.0 at 37°C and treated with or without 8 $\mu$g/mL (4 × MIC) POL, and with or without 2% glucose (Glc). At indicated time points (0 to 4 h), the number of live bacteria was determined by serial dilution and plating assay. *, $P < 0.05$; **, $P < 0.01$; ***, $P < 0.001$ by Student's *t* test.

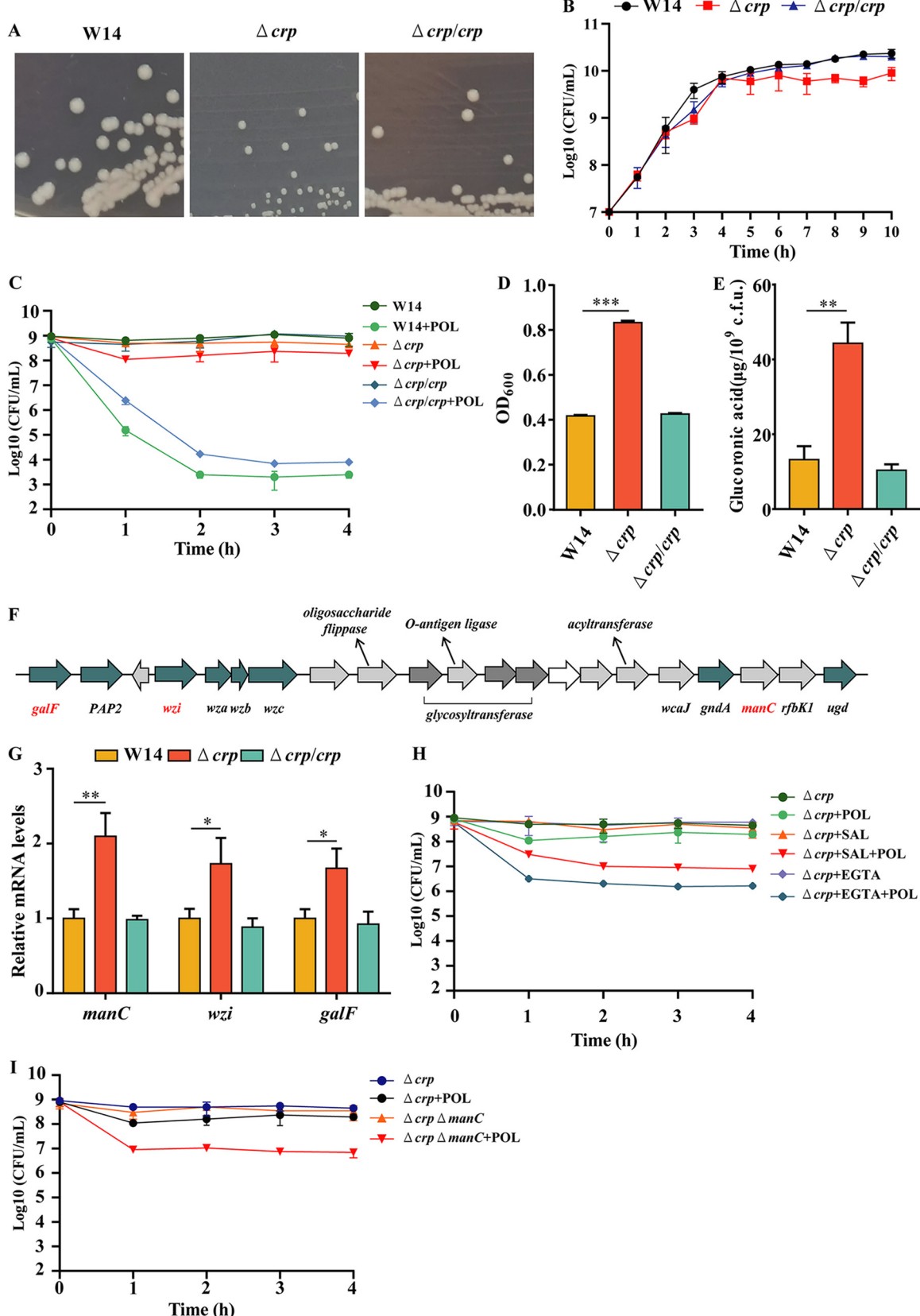

**FIG 2** CRP influenced bacterial resistance to polymyxins in HiAlc *Kpn*. Images of typical colonies (A) and growth curves (B) by WT, Δ*crp* mutant, and the complemented strain (Δ*crp/crp*). (C) WT, Δ*crp* mutant, and the complemented strain (Δ*crp/crp*) were grown to

**TABLE 2** Bacterial susceptibilities to polymyxins (standard inoculum: $1 \times 10^5$ CFU/well)

| Strain | MIC ($\mu$g /mL) | |
|---|---|---|
| | POL | COL |
| W14 | 2 | 2 |
| $\Delta crp$ | 8 | 8 |
| $\Delta crp/crp$ | 2 | 2 |

the glucose-induced polymyxins resistance in HiAlc *Kpn* (Fig. 3D). These results confirmed that glucose can affect the resistance of HiAlc *Kpn* against polymyxins by inhibiting the expression of *crp*. Notably, glucose did not increase the CPS content of $\Delta crp$ mutant strain (Fig. 3B and C), but significantly increased its survival rate under POL treatment (Fig. 3E). This suggests that glucose affected polymyxins resistance in HiAlc *Kpn* not only by increasing CPS production, but by other unknown pathways.

**Maintained ATP contributed to glucose-induced polymyxins resistance in HiAlc *Kpn*.** Glucose has been shown to induce polymyxins resistance via ATP maintenance in Gram-positive bacteria (10). However, it was unclear whether glucose could also induce polymyxins resistance in Gram-negative bacteria such as HiAlc *Kpn*. Nonlethal concentrations of POL significantly decreased intracellular ATP in the WT strain (Fig. 4A). This is most likely due to the ability of polymyxins to inhibit bacterial respiration via inhibition of type II NADH-quinone oxidoreductase (8). To clarify the role of ATP depletion in the bactericidal action of POL, the WT strain was pretreated with dicyclohexylcarbodiimide (DCCD), which is an inhibitor of F0-F1 ATP synthase (28). DCCD (1 mM) did not kill bacteria but significantly reduced the ATP content in the WT strain (Fig. 4B and C). WT strain pretreated with DCCD was easier to be killed by POL under high concentrations of glucose (Fig. 4B). Meanwhile, exogenous glucose could dramatically enhance the intracellular ATP level of POL-treated bacteria (Fig. 4A) and increased the bacterial survival rate (Fig. 1). In order to prove that the increased ATP caused by the addition of glucose can affect the resistance of bacteria to polymyxins, we treated HiAlc *Kpn* with 2.0 mM Na$_2$ATP, and the results showed that MIC (Table 1) and the survival rate (Fig. 4D) could be significantly increased. This is consistent with previous research results in Gram-positive bacteria—POL caused serious depletion of ATP and cell death through affecting the bacteria's respiration, and glucose promoted bacterial resistance to polymyxins by increasing the intracellular ATP (10).

**The inhibitors of TCA cycle and glycolysis attenuated glucose-induced polymyxins resistance in HiAlc *Kpn*.** The TCA cycle and glycolysis are important physiological processes affecting ATP production in bacteria. We treated the WT strain with malonic acid and cysteine, which are inhibitors of succinate dehydrogenase and pyruvate kinase, respectively, inhibiting the TCA flux and glycolysis, respectively (29, 30). Both cysteine and malonic acid significantly decreased the intracellular ATP and attenuated the glucose-induced polymyxins resistance in HiAlc *Kpn* (Fig. 5A and B). These results suggested that inhibition of ATP production could attenuate the glucose-induced polymyxins resistance in HiAlc *Kpn*, and suggested a possible new strategy for reducing glucose-induced polymyxins resistance.

**FIG 2** Legend (Continued)
an OD$_{600}$ of 0.8 to 1.0 at 37°C and treated with or without 8 $\mu$g/mL POL. At indicated time points (0 to 4 h), the number of live bacteria was determined by serial dilution and plating assay. (D and E) The mucoviscosity (D) and the amounts of capsular uronic acid (E) of WT, $\Delta crp$ mutant, and the complemented strain ($\Delta crp/crp$). (F) Schematic of the capsule locus in W14. Genes in green are highly conserved among different capsule types. (G) Expression levels of CPS biosynthesis genes in WT, $\Delta crp$ mutant, and the complemented strain ($\Delta crp/crp$) were grown to an OD$_{600}$ of 0.8 to 1.0 at 37°C, followed by RNA extraction. The mRNA levels of *galF*, *wzi*, and *manC* were determined by real-time PCR with *rpoB* as the internal control. (H and I) The $\Delta crp$ mutant strain was grown to an OD$_{600}$ of 0.8 to 1.0 with or without 30 $\mu$g/mL sodium salicylate (SAL) or 1 mM EGTA and treated with or without 8 $\mu$g/mL POL (H). The $\Delta crp$ and $\Delta crp\Delta manC$ mutant strains were grown to an OD$_{600}$ of 0.8 to 1.0 and treated with or without 8 $\mu$g/mL POL (I). At indicated time points (0 to 4 h), the number of live bacteria was determined by serial dilution and plating assay. *, $P < 0.05$; **, $P < 0.01$; ***, $P < 0.001$ by Student's *t* test.

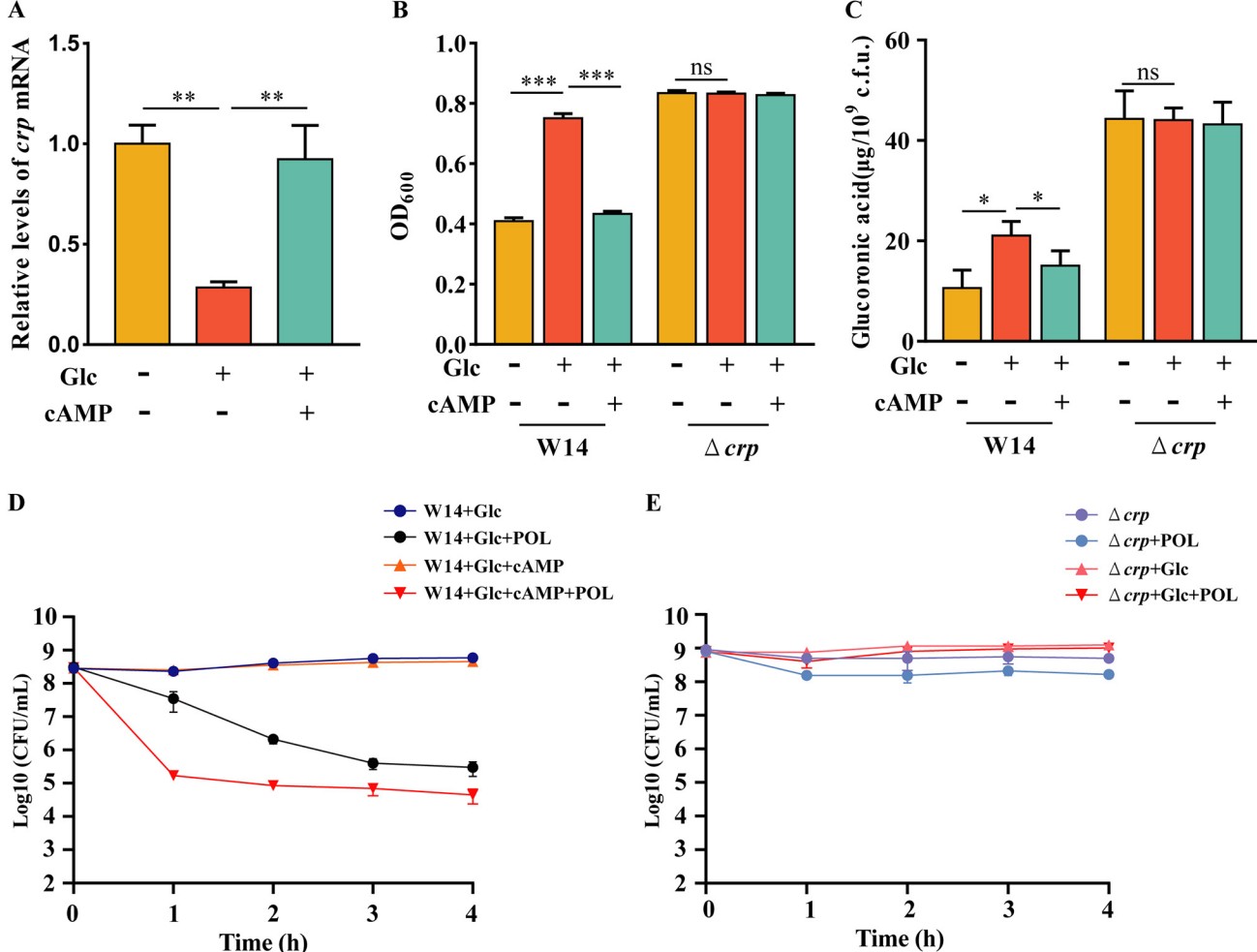

**FIG 3** Downregulation of *crp* contributed to glucose-induced polymyxins resistance in HiAlc *Kpn*. (A) WT, WT plus 2% Glc, and WT plus 2% Glc and 1 mM cAMP strains were grown to an OD$_{600}$ of 0.8 to 1.0 at 37°C, followed by RNA extraction. The mRNA levels of *crp* were determined by real-time PCR with *rpoB* as the internal control. (B and C) The mucoviscosity (B) and the amounts of capsular uronic acid (C) of WT and Δ*crp* mutant strains after treatment of POL, POL plus 2% Glc and POL plus 2% Glc and 1 mM cAMP. (D) Strains were treated by 2% Glc, 2% Glc plus POL, 2% Glc plus 1 mM cAMP, and 2% Glc plus 1 mM cAMP and POL. (E) Strains were treated by POL, 2% Glc and 2% Glc plus POL. At indicated time points (0 to 4 h), the number of live bacteria was determined by serial dilution and plating assay. **, $P < 0.01$; ***, $P < 0.001$ by Student's *t* test.

## DISCUSSION

High-sugar diet can disrupt immune-mediated protection from metabolic syndrome by causing microbiota imbalance, which is harmful to health (31). Previously, we found that when patients with severe NAFLD accompanied by auto-brewery syndrome ate a high-sugar diet, the HiAlc *Kpn* located in the patients' gut produced excess endogenous alcohol from sugar, and glucose was often the sugar they used first (32). In the present study, we found that glucose could increase the resistance of HiAlc *Kpn* to polymyxins by increasing the CPS and maintaining intracellular ATP (Fig. 6).

As the last line of antibiotic defense, polymyxins is often used to treat multidrug-resistant *K. pneumoniae* strains, especially CRKP (5). When *K. pneumoniae* was exposed to high concentrations of glucose, the increased CPS not only protected bacteria from killing by phagocytosis or serum factors but also increases the bacterial resistance to polymyxins (33, 34). This undoubtedly increases the difficulty to clinically treat HiAlc *Kpn* infections. Our present study found that inhibition of CPS formation by addition of cAMP, and inhibition of energy metabolic processes by inhibition of the TCA cycle or glycolysis, significantly enhanced the bactericidal ability of polymyxins. This will help to improve the efficacy of polymyxins in treating HiAlc *Kpn* infections. Meanwhile, we sought to investigate whether the glucose-induced polymyxins resistance mechanism

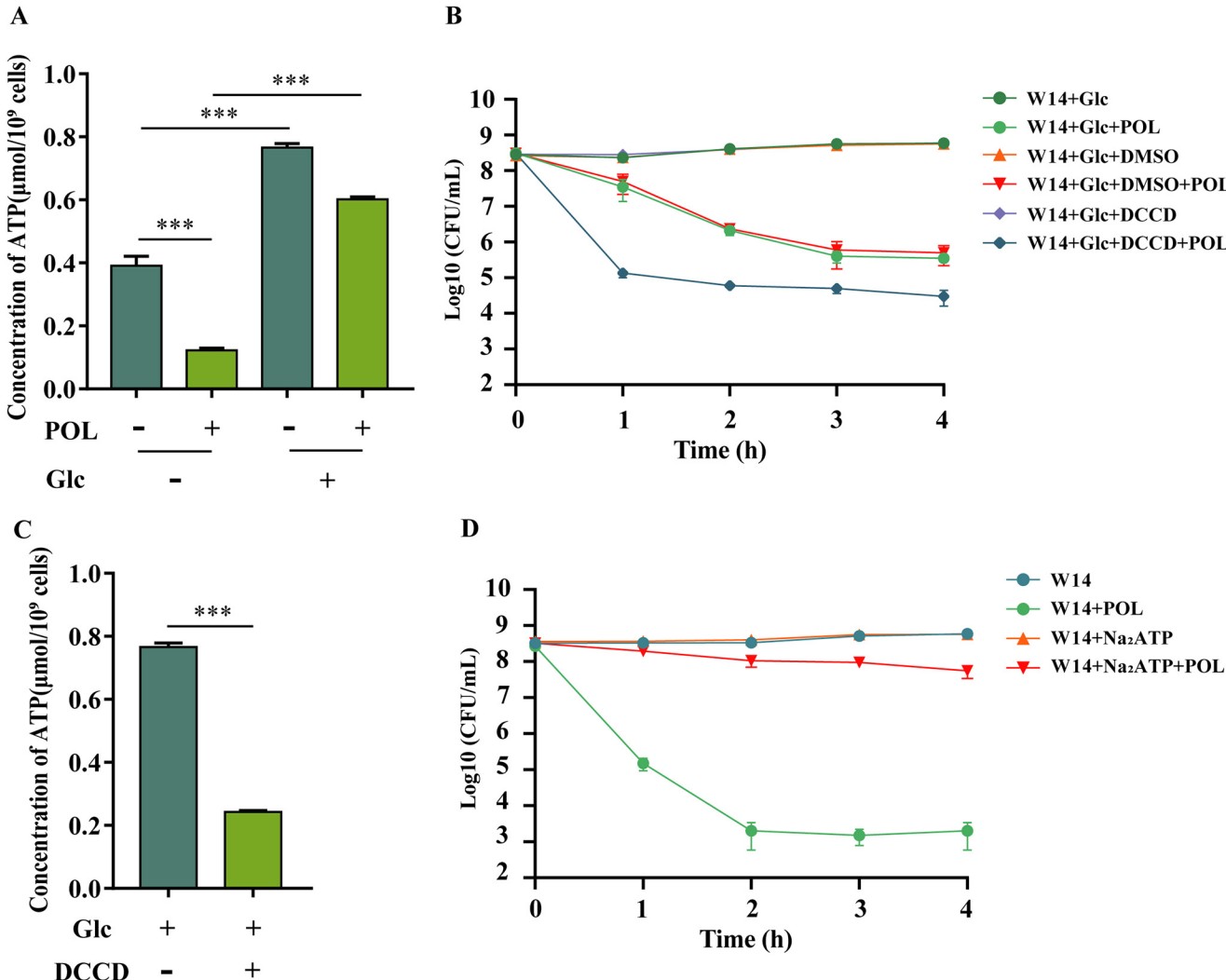

**FIG 4** Maintained ATP contributed to glucose-induced polymyxins resistance in HiAlc *Kpn*. (A) *In vivo* ATP levels of WT strain after no treatment and treatment of POL, Glc, and POL plus Glc. (B) The survival rates of WT strain in 2% Glc after treatment of POL, DMSO, POL plus DMSO, 1.0 mM DCCD, and POL plus 1.0 mM DCCD. DCCD was dissolved in DMSO. At indicated time points (0 to 4 h), the number of live bacteria was determined by serial dilution and plating assay. (C) *In vivo* ATP levels of WT strain after treatment of Glc and Glc plus 1.0 mM DCCD. **, $P < 0.01$; ***, $P < 0.001$ by Student's *t* test. (D) The survival rates of WT strain in 2% Glc after treatment of POL, 2.0 mM Na$_2$ATP, and POL plus 2.0 mM Na$_2$ATP. At indicated time points (0 to 4 h), the number of live bacteria was determined by serial dilution and plating assay.

was limited to HiAlc *Kpn*. The MIC and survival rate of low-alcohol-producing *K. pneumoniae* (LowAlc *Kpn*) under POL treatment in the presence 2% glucose were determined. When treated by glucose, LowAlc *Kpn* showed a higher-level resistance (MICs) to POL (Table 1). Consistent with the MIC test results, the survival rate of LowAlc *Kpn* also increased significantly in the presence of glucose. And addition of cAMP or consumption of intracellular ATP by DCCD both could significantly inhibit the glucose-induced polymyxins resistance in LowAlc *Kpn* (Data not shown). These results showed that glucose-induced polymyxins resistance was common in *K. pneumoniae*. In Gram-positive bacteria, polymyxins could kill bacteria by depleting intracellular ATP, and it is glycolysis, not the TCA cycle, that plays an important role in glucose-induced polymyxins resistance (10). However, in this study, both glycolysis and TCA cycle played an important role in glucose-induced polymyxins resistance. The reasons for this difference still need to be further explored.

Several regulators have been confirmed to be involved in the regulation of polymyxins resistance in *K. pneumoniae*, including PhoP/PhoQ, PmrA/PmrB, and CrrA/CrrB, whose main mechanism of action is to modify the outer membrane of bacteria (35,

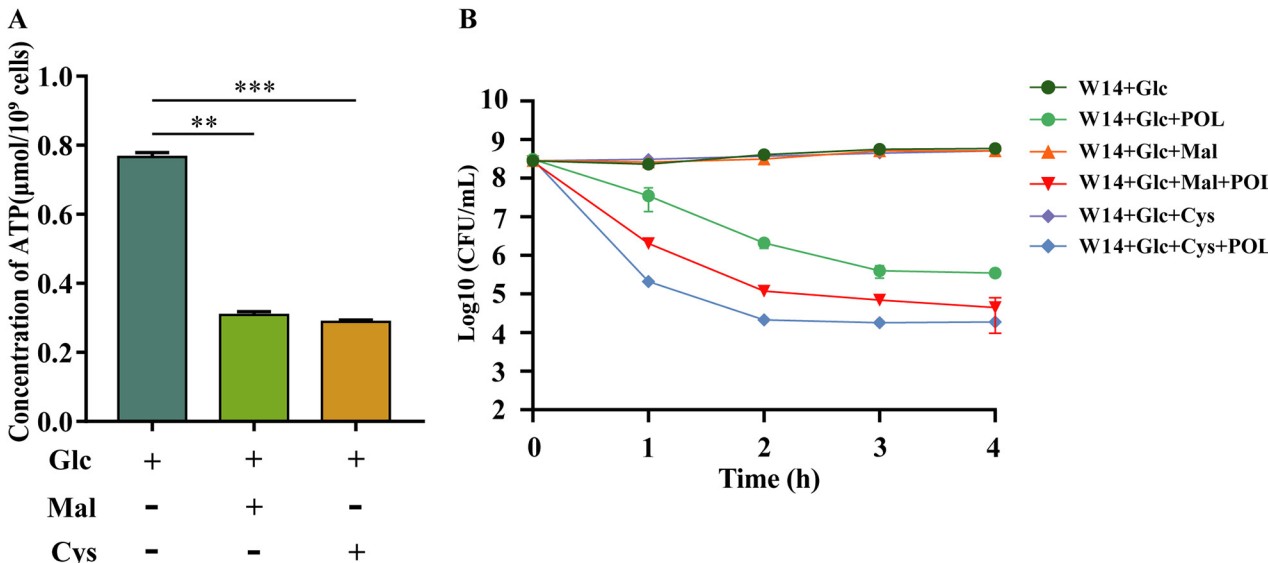

**FIG 5** Effect of inhibitors on glucose-induced polymyxins resistance in HiAlc *Kpn*. (A) *In vivo* ATP levels of WT strain after treatment of Glc, Glc plus 10 mM cysteine (Cys), and Glc plus 20 mM malonic acid (Mal). (B) The survival rates of WT strain in 2% Glc after treatment of POL, Cys, POL plus Cys, Mal, and POL plus Mal. At indicated time points (0 to 4 h), the number of live bacteria was determined by serial dilution and plating assay. **, $P < 0.01$; ***, $P < 0.001$ by Student's *t* test.

36). As a global regulator, CRP regulates many aspects of bacteria, such as carbon catabolite repression, pathogenicity, biofilm formation, cell morphology, fimbria production, and capsular polysaccharide biosynthesis (24, 25). In addition, CRP can also affect bacterial antibiotic resistance. In *E. coli*, cAMP/CRP plays an important role in persister metabolism, as its perturbation can effectively eliminate the ability of metabolites to enable aminoglycoside killing of persister (37). In this study, we revealed that CRP could affect polymyxins resistance of HiAlc *Kpn* by regulating the CPS. However, either inhibition of capsule formation by treating bacteria with EAL and EGTA, or deletion of major gene *manC* involved in capsule formation can only partially compensate for drug-resistant phenotypes. This suggested that CRP may affect polymyxins resistance in HiAlc *Kpn* through other regulatory mechanisms. Notably, the growth rate of Δ*crp* mutant was slower. Whether there is a relationship between the reduced growth rate of Δ*crp* and the increased resistance to polymyxins is worth further investigation.

In conclusion, we found that glucose could induce the polymyxins resistance in HiAlc *Kpn* via increasing capsular polysaccharide and maintaining intracellular ATP, further deepening the understanding of its mechanism of polymyxins resistance.

## MATERIALS AND METHODS

**Bacterial strains, plasmids, and primers.** *K. pneumoniae* W14 (WT), a high-alcohol-producing strain, was originally isolated from a patient with severe NAFLD accompanied by auto-brewery syndrome (1). For general bacterial cultivation and phenotypic assays, all strains were cultured in Luria-Bertani (LB) broth (5 g/L yeast extract, 10 g/L sodium chloride, and 10 g/L tryptone) at 37°C with agitation at 200 rpm. All the strains, the plasmids and the primers related to this study are listed in Table S1.

**Mutant and complementation construction.** To obtain the gene deletion or gene complementation strains, methods by Link et al. and Pan et al. were used (38, 39). For the construction of the Δ*crp* mutant strain, the flanking regions of *crp* were ligated into a NotI-digested pKO3-Km plasmid using a one-step cloning method. The resulting plasmid was transformed into W14 using electroporation. At 43°C, the plasmid was integrated into the bacterial chromosome, and the positive colonies were grown at 30°C. Wild-type or mutant strains appeared on the plate with sucrose, which was confirmed via PCR. For the construction of the *crp* complementation strain, *crp* and its ribosomal binding site were cloned into a pGEM-T Easy plasmid, which was digested by SalI and sphI. The resulting plasmid was transformed into Δ*crp* mutant strain using electroporation.

**RNA extraction, reverse transcription, and quantitative PCR.** The bacterial solution to be tested was centrifuged at $12,000 \times g$, and the bacterial precipitate was resuspended with 1 mL TRIzol. Following this, 0.2 mL of trichloromethane was added, vortexed for 15 s, and centrifuged at $12,000 \times g$ for 10 min. An equal volume of aqueous phase was slightly mixed with isopropanol, and the total RNA

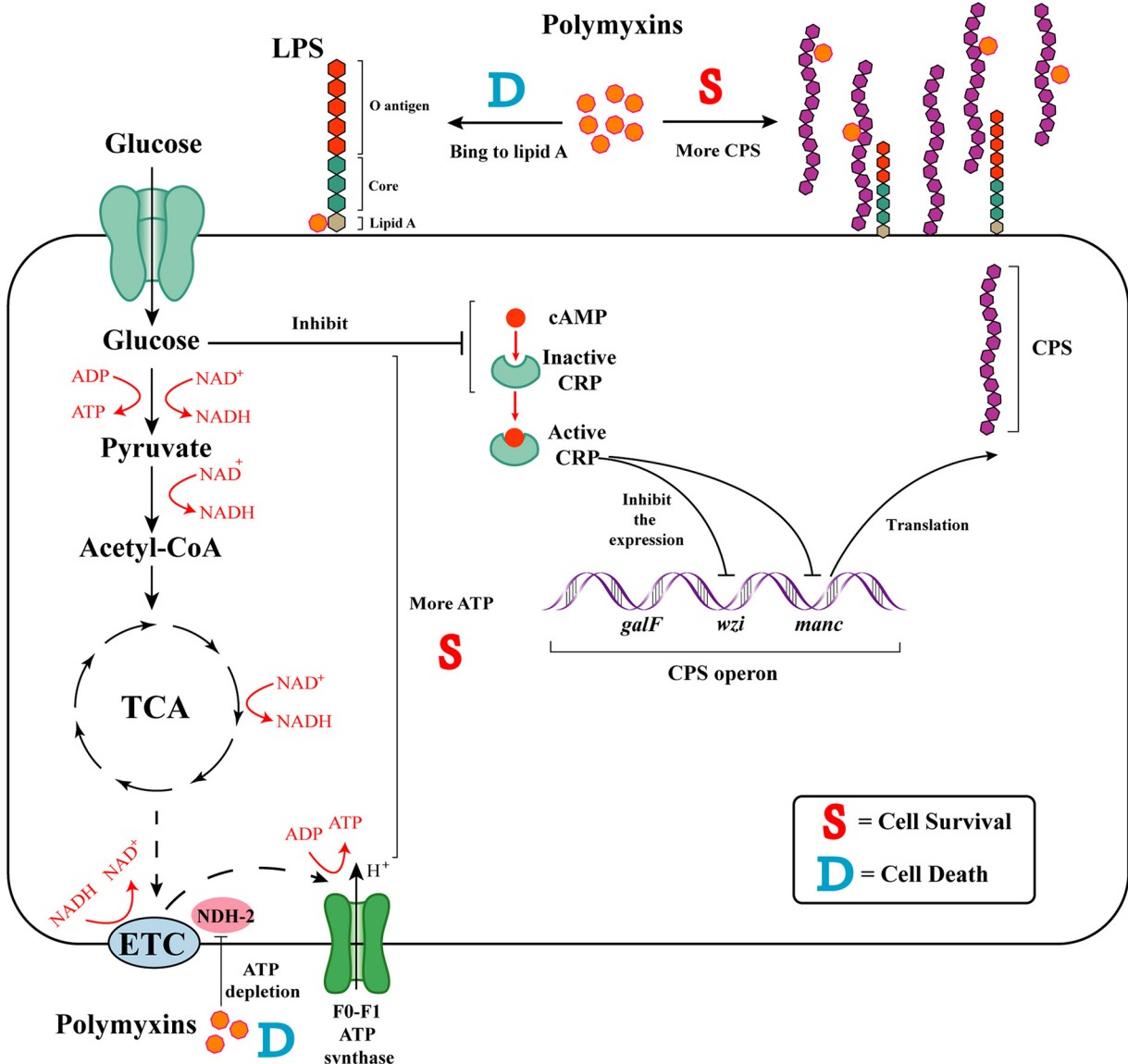

**FIG 6** Proposed model of glucose-induced resistance to polymyxins in HiAlc *Kpn*. Polymyxins can electrostatically bind to lipid A or inhibit the type II NADH-quinone oxidoreductase (NDH-2), leading to the leakage of cytoplasmatic contents and ATP depletion, respectively, contribute to cell death. Glucose induces bacterial resistance to polymyxins in two ways. First, glucose inhibits the expression of *crp*, promotes the production of CPS, and then induces the resistance of HiAlc *Kpn* to polymyxins. Second, glucose can maintain high ATP level by tricarboxylic acid (TCA), glycolysis and electron transport chain (ETC), effectively increasing bacterial survival.

was then extracted with the help of a RNeasy minikit (Tiangen Biotech, Beijing, China). Approximately 0.5 to 1 $\mu$g of RNA was used for reverse transcription, and cDNA was synthesized using PrimeScript Reverse Transcriptase (TaKaRa, Dalian, China). The quantitative PCR was operated on the CFX Connect Real-Time system (Bio-Rad, USA) in a 20 $\mu$L mix, including the SYBR Premix Ex TaqTM II (TaKaRa, Dalian, China), cDNA, and specific forward and reverse primers. *rpoB*, which encodes the DNA-directed RNA polymerase subunit beta, was used as an internal control.

**MICs and survival assay.** The MICs were determined using the 2-fold broth microdilution method in accordance with Clinical and Laboratory Standards Institute (CLSI) M07 11th edition and M100 32th edition. *E. coli* ATCC 25922 was used as quality control. The 96-well plates were incubated at 37°C for 20 h. The smallest concentration of antibiotic that caused bacterial growth to be invisible to the naked eye was recorded as the MIC. To test bacterial survival rate after treatment with polymyxins, overnight cultures of bacteria were transferred to fresh LB broth at a ratio of 1:100. When the OD600 reached 0.8 to 1.0, the bacteria were mixed with the indicated concentration of polymyxins in the 96-well plates. The 96-well plates were incubated at 37°C without agitation. At indicated time points (0 to 4 h), the number of live bacteria was determined by plating.

**Mucoviscosity assay.** To measure the levels of mucoviscosity in *K. pneumoniae*, low-speed centrifugation was performed as previously described (40). Briefly, equal numbers of overnight-cultured bacteria

were centrifuged at 1,500 $\times$ *g* for 5 min, and the absorbance of the supernatant was measured at 600 nm.

**Uronic acid measurement.** Uronic acid is a key component of many capsules and an indicator of capsule content (40). To measure the capsule content, the hot phenol-water method was used to extract CPS from *K. pneumoniae* (41). Bacterial samples to be tested were collected via centrifugation at 12,000 $\times$ *g* and resuspended in 0.5 mL distilled water. Following this, the samples were incubated at 68℃ for 2 min, and the incubation was continued for another 30 min after adding 0.5 mL phenol. After the sample was restored to room temperature, trichloromethane was added, and the aqueous phase was obtained through centrifugation at 12,000 $\times$ *g*. CPS was then obtained through ethanol precipitation at $-$20℃. A modified carbazole assay was used to quantify the uronic acid content (41). Briefly, 0.1 mL purified CPS was mixed with 0.6 mL 0.025 M sodium tetraborate dissolved in sulfuric acid, and the sample was then heated to 100℃ for 15 min. After the sample was restored to room temperature, 0.02 mL of 0.125% (wt/vol) carbazole dissolved in absolute ethanol was added, and then the sample was heated for another 15 min. The sample was cooled, and $OD_{530}$ was determined. A standard curve was generated with glucuronic acid.

**Determination of ATP level.** Enhanced ATP assay kit (Beyotime Biotechnology, Shanghai, China) was used to detect the ATP level. Briefly, the bacterial samples were centrifuged at 12,000 $\times$ *g*, and the bacterial precipitate was resuspended with 1 $\times$ PBS buffer and washed one to two times. The ATP detection reagent was mixed with the diluent at a ratio of 1:4 and configured as ATP working solution. A total of 0.02 mL of the sample to be tested was mixed with 0.1 mL ATP working solution, and the relative light units (RLU) value was detected using a luminometer and compared with a standard curve generated using ATP standard solution.

**Statistical analysis.** Three biological replicates were used for each assay. Data were statistically analyzed using GraphPad Prism (version 5.0, USA) and presented as the mean $\pm$ standard deviations (SD). Student's *t* test was performed to determine statistically significant differences, and *, $P < 0.05$; **, $P < 0.01$; and ***, $P < 0.001$ all indicated statistical significance.

## SUPPLEMENTAL MATERIAL

Supplemental material is available online only.

**SUPPLEMENTAL FILE 1**, PDF file, 0.2 MB.

## ACKNOWLEDGMENTS

We thank Editage (www.editage.cn) for English language editing. This work was supported by grants from the National Natural Science Foundation for Key Programs of China Grants (82130065), National Natural Science Foundation of China (32200159, 32170201 and 82002191), Beijing Natural Science Foundation (7222014), FENG foundation (FFBR 202103), the Research Foundation of Capital Institute of Pediatrics (CXYJ-2021-04, JCYJ-2023-02, and JCYJ-2023-09), Public service development and reform pilot project of the Beijing Medical Research Institute (BMR2019-11). Postdoctoral Research Fund of Chaoyang District, Beijing, China in 2021.

J.Y., Z.F., and T.F. designed the experiments; Z.F., T.F., H.L., Z.L., B.D., X.C., R.Z., Y.F., and H.Z. performed the experiments; all other authors analyzed the results; Z.F. and T.F. wrote the manuscript; Y.J. and G.X. revised the manuscript. All authors reviewed the manuscript.

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
