## [Reviewer comments · Microbiology Spectrum]

Microbiology Spectrum

Glucose Induces Resistance to Polymyxins in High-alcohol-producing *Klebsiella pneumoniae* via Increasing Capsular Polysaccharide and Maintaining Intracellular ATP

Zheng Fan, Tongtong Fu, Hongbo Liu, Zhoufei Li, Bing Du, Xiaohu Cui, Rui Zhang, Yanling Feng, Hanqing Zhao, Guanhua Xue, Jinghua Cui, Chao Yan, Lin Gan, Junxia Feng, Ziyang Xu, Zihui Yu, Ziyang Tian, Zanbo Ding, Jinfeng Chen, Yujie Chen, and Jing Yuan

Corresponding Author(s): Jing Yuan, Capital Institute of Pediatrics

Review Timeline:

Submission Date:	January 10, 2023
Editorial Decision:	February 26, 2023
Revision Received:	March 29, 2023
Editorial Decision:	May 1, 2023
Revision Received:	May 8, 2023
Accepted:	May 31, 2023

Editor: Hui Wang

Reviewer(s): Disclosure of reviewer identity is with reference to reviewer comments included in decision letter(s). The following individuals involved in review of your submission have agreed to reveal their identity: Yunsong Yu (Reviewer #2)

Transaction Report:

DOI: <https://doi.org/10.1128/spectrum.00031-23>

February 26, 2023

Prof. Jing Yuan
Capital Institute of Pediatrics
Department of Bacteriology
No. 2 yabao road, Chaoyang District
Beijing, Beijing 100020
China

Re: Spectrum00031-23 (Glucose Induces Resistance to Polymyxins in High-alcohol-producing *Klebsiella pneumoniae* via Increasing Capsular Polysaccharide and Maintaining Intracellular ATP)

Dear Prof. Jing Yuan:

Link Not Available

Sincerely,

Hui Wang

Journals Department
Reviewer comments:

Reviewer #1 (Comments for the Author):

summary

In this paper, the authors assessed the role of glucose in polymyxin resistance of HiA1c *K. pneumoniae*. They found that the repression of *crp* and maintenance of ATP production were induced by the addition of glucose and hence lower the susceptibility to polymyxin. It is no surprise that glucose, as a core carbon source, involved in capsule production and ATP supply that eventually result in altered antimicrobial susceptibility. It seems this mechanism is not exclusive to the HiA1c *K. pneumoniae*. It is not necessary to emphasize the resistance mechanism found in HiA1c strain. The paper is generally well written and the points

are easy to follow, but the novelty and importance of the conclusions should be further elucidated.

Major concern

1. L136-142, Figure 1-5, the survival rates were calculated by the number of live cells before and after antibiotic treatment. This method cannot rule out the effect of growth rate on the number of surviving bacteria, especially for the *crp* and *manC* mutant. The growth control of no antibiotic treatment should also be performed plate counting after 1-hour culture. The evaluation of the resistance to antibiotic cannot only detect the 1-hour survival rate. A growth curve based on viable counting should be conducted.

2. L130-136, Table 2, AST and interpretation should refer to the standard methods described in CLSI documents (M07 and M100). Cation-adjusted Mueller-Hinton broth not the LB medium was used for susceptibility testing. That is to say, the 'MIC' determined by the nutrient supplemented medium different from that by the standard method is reasonable. The standard incubation conditions (35°C±2°C for 16-20 hours) may not suitable for the richer medium with glucose addition. The authors should estimate the effect of glucose on the susceptibility in other strains, including the strain for quality control.

Specific points

1. Move the Table 1 to supplement
2. Table 2, the MIC from 2 to 4 is not a significant change. Quality control should be included.
3. Figure 2, cps production was significantly increased in *crp* mutant, but the colony morphology can not reflect that.
4. figure 4, why the DMSO was compared

Reviewer #2 (Comments for the Author):

The study found a novel mechanism of polymyxins resistance, via increasing capsular polysaccharide and maintaining intracellular ATP. This study was well-designed, and the experiment was completed. I think it is very interesting, and with minor revision, I think this manuscript could be published.

Comments :

1. The CRP is a global regulator, and the growth rate of *crp* knockout isolate was slower. And growth rate of isolates when adding some inhibitors should be determined. The relationship between growth rate and polymyxins resistance should be discussed. Further, the transcriptome analysis could be performed, if possible.
2. Figure 2B: As the mucoviscosity of the isolates was different, performing a growth curve using OD method is not appropriate. I recommend the authors perform it by counting CFU.
3. Line 46: Inflammatory bowel diseases and nonalcoholic fatty liver diseases should not be considered as infectious diseases.
4. Line 61: *mcr* is a kind of LPS modification.

Staff Comments:

Preparing Revision Guidelines

Please return the manuscript within 60 days; if you cannot complete the modification within this time period, please contact me. If you do not wish to modify the manuscript and prefer to submit it to another journal, please notify me of your decision immediately so that the manuscript may be formally withdrawn from consideration by Microbiology Spectrum.

If your manuscript is accepted for publication, you will be contacted separately about payment when the proofs are issued; please follow the instructions in that e-mail. Arrangements for payment must be made before your article is published. For a

complete list of **Publication Fees**, including supplemental material costs, please visit our website.

Dear Microbiology spectrum editors and reviewers,

We appreciate the time and efforts by the editor and reviewers of 'Microbiology spectrum' in reviewing this manuscript. In response to the reviewer's remarks, we have revised our manuscript meticulously to our capacity and tried to incorporate all the suggestions made by the reviewers. We hope that the revised version could meet the publication requirements of Microbiology spectrum. And the modified/corrected parts are mentioned according to the line numbers.

A point-by-point response to Reviewer #1 Comments:

Summary

In this paper, the authors assessed the role of glucose in polymyxin resistance of HiAlc *K. pneumoniae*. They found that the repression of *crp* and maintenance of ATP production were induced by the addition of glucose and hence lower the susceptibility to polymyxin. It is no surprise that glucose, as a core carbon source, involved in capsule production and ATP supply that eventually result in altered antimicrobial susceptibility. It seems this mechanism is not exclusive to the HiAlc *K. pneumoniae*. It is not necessary to emphasize the resistance mechanism found in HiAlc strain. The paper is generally well written and the points are easy to follow, but the novelty and importance of the conclusions should be further elucidated.

Response:

Thank you very much for your suggestions, which are very important to us. This work was carried out based on our previous findings that high-alcohol-producing *K. pneumoniae* (HiAlc *Kpn*) is one of the main causes of nonalcoholic fatty liver disease (NAFLD). HiAlc *Kpn* could produce excess endogenous alcohol in the gut of patients with NAFLD, using glucose as the main carbon source (1). Glucose affects the resistance of HiAlc *Kpn* to polymyxins, which may affect the therapeutic effect of polymyxins on NAFLD caused by HiAlc *Kpn* with polymyxins resistance. This is the

reason why we emphasize the resistance mechanism found in HiAlc *Kpn*. Based on your valuable suggestion, we examined the survival rate of low-alcohol-producing *K. pneumoniae* (LowAlc *Kpn*) treated by polymyxins in the presence or absence of glucose. The results showed that glucose enhanced the survival rate of LowAlc *Kpn* under polymyxins treatment. Meanwhile, addition of cAMP or consumption of intracellular ATP by DCCD both could significantly inhibit the glucose-induced polymyxins resistance. Hence, in the discussion section, we emphasized that glucose-induced polymyxins resistance is common in *K. pneumoniae*. Please refer to line 276-283 for the changes. However, we have retained the description of HiAlc *Kpn* on the basis that 'high-alcohol-producing' is a special characteristic of HiAlc *Kpn*, which is different from other *K. pneumoniae* and is one of the main causes of NAFLD.

Major concern

1. L136-142, Figure 1-5, the survival rates were calculated by the number of live cells before and after antibiotic treatment. This method cannot rule out the effect of growth rate on the number of surviving bacteria, especially for the *crp* and *manC* mutant. The growth control of no antibiotic treatment should also be performed plate counting after 1-hour culture. The evaluation of the resistance to antibiotic cannot only detect the 1-hour survival rate. A growth curve based on viable counting should be conducted.

Response:

Thank you very much for your suggestions. All survival assay results have been changed to time-dependent growth curves based on viable counting. And all growth control of no antibiotic treatment were added. Please refer to Figure 1-5.

2. L130-136, Table 2, AST and interpretation should refer to the standard methods described in CLSI documents (M07 and M100). Cation-adjusted Mueller-Hinton broth not the LB medium was used for susceptibility testing. That is to say, the 'MIC' determined by the nutrient supplemented medium different from that by the standard method is reasonable. The standard

incubation conditions (35°C{plus minus}2°C for 16-20 hours) may not suitable for the richer medium with glucose addition. The authors should estimate the effect of glucose on the susceptibility in other strains, including the strain for quality control.

Response:

Thank you very much for your suggestions.

i) We have re-conducted the minimum inhibitory concentrations (MICs) assays with reference to the standard methods described in CLSI documents (M07 11th edition and M100 32th edition). Please refer to Table 1, Table 2 and line 126-136 for the changes. And we modified the term 'Susceptible' in Figure 6, which is not appropriate to describe polymyxins resistance in bacteria.

ii) We quite agree with you that the standard incubation conditions (35°C{plus minus}2°C for 16-20 hours) may not suitable for the richer medium with glucose addition. Therefore, we combined MIC assays with bacterial survival rate calculation to reflect the effect of glucose on bacterial resistance. We added an assessment of the effect of glucose on drug resistance in low-alcohol-producing *Klebsiella pneumoniae* strains and the quality control strain *E.coli* ATCC 25922. Glucose can cause a 2-4 fold increase in the MIC of these strains to polymyxins. In order to prove that the increased ATP caused by the addition of glucose can affect the resistance of bacteria to polymyxins, we treated bacteria with 2.0 mM Na₂ATP, and the results showed that MIC and the survival rate of bacteria could be significantly increased. Please refer to Table 1 and Figure 4D for the changes.

Specific points

1. Move the Table 1 to supplement

Response:

Thanks for your suggestion. We have already moved the Table 1 to the Supplemental Material section and named it Supplementary Table S1.

2. Table 2, the MIC from 2 to 4 is not a significant change. Quality control should

be included.

Response:

Thanks for your suggestion. We repeated the experiment for three times, and everytime the MIC was from 2 to 4. The same experimental phenomenon occurred with the quality control strain *E.coli* ATCC 25922. You're right, the MIC from 2 to 4 is not a significant change. But by calculating bacterial survival rate, the effect of glucose on bacterial resistance was significant. In order to prove that the increased ATP caused by the addition of glucose can affect the resistance of bacteria to polymyxins, we treated bacteria with Na_2ATP , and the results showed that MIC and the survival rate of bacteria could be significantly increased. Please refer to Table 1 and Figure 4D for the changes.

3. Figure 2, cps production was significantly increased in *crp* mutant, but the colony morphology can not reflect that.

Response:

Thanks for your suggestion. Deletion of *crp* resulted in slower bacterial growth in both solid and liquid cultures, which may be one reason why the colony morphology did not show an increased CPS. This has also been mentioned in previous studies, " After 16 h incubation on the blood agar, Δcrp formed colonies at a diameter of 1.0 ± 0.1 mm, which was much smaller than that of WT (4.0 ± 0.5 mm). " (2) Although colony morphology does not show an increase in the CPS well, we assessed the mucoviscosity of the capsule by low-speed centrifugation of liquid cultures as shown in the figure below (3), WT formed more dense pellets than Δcrp when equal numbers of overnight-cultured bacteria were centrifuged.

4. Figure 4, why the DMSO was compared

Response:

Thanks for your suggestion. Since DCCD was dissolved in DMSO, we also examined the effect of DMSO on bacterial survival. We have added relevant explanations in the results description section. Please refer to line 505-506 for the changes.

A point-by-point response to Reviewer #2 Comments:

The study found a novel mechanism of polymyxins resistance, via increasing capsular polysaccharide and maintaining intracellular ATP. This study was well-designed, and the experiment was completed. I think it is very interesting, and with minor revision, I think this manuscript could be published.

Comments:

1. The CRP is a global regulator, and the growth rate of *crp* knockout isolate was slower. And growth rate of isolates when adding some inhibitors should be determined. The relationship between growth rate and polymyxins resistance should be discussed. Further, the transcriptome analysis could be performed, if possible.

Response:

We thank for the reviewer's constructive suggestion, which are very important to us.

i) We added the determination of the growth rate of strains when adding the inhibitors of TCA cycle and glycolysis. Please refer to Figure 4B and Figure 5B.

ii) It was not clear whether there was a relationship between the reduced growth rate of Δcrp strain and the resistance to polymyxins. In our another study, deletion of regulator IHF also resulted in a decrease in bacterial growth, but increased susceptibility to polymyxins (Manuscript in preparation). As a global regulator, CRP regulates many aspects of bacteria, and bacterial growth rate and antibiotic resistance are also determined by many factors. In the discussion section, we added a description of the relationship between bacterial growth rate and resistance to polymyxins. Please refer to line 303-305 for the changes.

iii) In order to explore whether CRP affect the resistance of bacteria to polymyxins

through other mechanisms besides the CPS, we performed the transcriptome analysis of WT and Δcrp (China National Center for Bioinformation, accession number CRA010408). However, no genes known to influence polymyxins resistance, such as *phoP/phoQ*, *pmrA/pmrB*, *acrA/acrB*, were found in differentially expressed genes analysis. Potential new regulation mechanism need further study.

2. Figure 2B: As the mucoviscosity of the isolates was different, performing a growth curve using OD method is not appropriate. I recommend the authors perform it by counting CFU.

Response:

As reviewer's suggestion, we have already performed the growth curve by counting CFU. Please refer to Figure 2B.

3. Line 46: Inflammatory bowel diseases and nonalcoholic fatty liver diseases should not be considered as infectious diseases.

Response:

Thanks for your comment. We have already revised the sentence to " *Klebsiella pneumoniae* is an opportunistic bacterial pathogen that causes various infections, including urinary tract infections, hospital-acquired pneumonia (HAP), bacteremia and liver abscesses ". Please refer to line 44-46 for the changes.

4. Line 61: *mcr* is a kind of LPS modification.

Response:

Thanks for the reviewer's comments and correction. We are sorry about this error. The *mcr* genes encode phosphoethanolamine (pEtN) transferase enzymes, which can reduce the electrostatic interaction between polymyxins and lipid A of LPS by binding pEtN moiety to the lipid A, creating bacteria resistant to polymyxins. So we deleted this sentence.

REFERENCES

1. Yuan J, Chen C, Cui J, Lu J, Yan C, Wei X, Zhao X, Li N, Li S, Xue G, Cheng W, Li B, Li H, Lin W, Tian C, Zhao J, Han J, An D, Zhang Q, Wei H, Zheng M, Ma X, Li W, Chen X, Zhang Z, Zeng H, Ying S, Wu J, Yang R, Liu D. 2019. Fatty Liver Disease Caused by High-Alcohol-Producing *Klebsiella pneumoniae*. *Cell Metab* 30:675-688. 10.1016/j.cmet.2019.08.018.
2. Ou Q, Fan J, Duan D, Xu L, Wang J, Zhou D, Yang H, Li B. 2017. Involvement of cAMP receptor protein in biofilm formation, fimbria production, capsular polysaccharide biosynthesis and lethality in mouse of *Klebsiella pneumoniae* serotype K1 causing pyogenic liver abscess. *J Med Microbiol* 66:1-7. 10.1099/jmm.0.000391.
3. Walker KA, Miner TA, Palacios M, Trzilova D, Frederick DR, Broberg CA, Sepulveda VE, Quinn JD, Miller VL. 2019. A *Klebsiella pneumoniae* Regulatory Mutant Has Reduced Capsule Expression but Retains Hypermucoviscosity. *mBio* 10. 10.1128/mBio.00089-19.

May 1, 2023

Prof. Jing Yuan
Capital Institute of Pediatrics
Department of Bacteriology
No. 2 yabao road, Chaoyang District
Beijing, Beijing 100020
China

Re: Spectrum00031-23R1 (Glucose Induces Resistance to Polymyxins in High-alcohol-producing *Klebsiella pneumoniae* via Increasing Capsular Polysaccharide and Maintaining Intracellular ATP)

Dear Prof. Jing Yuan:

Link Not Available

Sincerely,

Hui Wang

Journals Department
Reviewer comments:

Reviewer #1 (Comments for the Author):

The author has modified the experimental method and the results shown in the tables and figures, but the corresponding text has not been modified, such as L178, there is actually no COL in Table 1, and the LowAlc Kpn strain are not mentioned in the maintext.

I have nothing else to point out.

Reviewer #2 (Comments for the Author):

The manuscript is well-written and suggested to be published without revision.

Staff Comments:

Preparing Revision Guidelines

Please return the manuscript within 60 days; if you cannot complete the modification within this time period, please contact me. If you do not wish to modify the manuscript and prefer to submit it to another journal, please notify me of your decision immediately so that the manuscript may be formally withdrawn from consideration by Microbiology Spectrum.

Dear Microbiology spectrum editors and reviewers,

We appreciate the time and efforts by the editor and reviewers of “Microbiology spectrum” in reviewing this manuscript. In response to the reviewer's remarks, we have revised our manuscript meticulously to our capacity and tried to incorporate all the suggestions made by the reviewers. The modified/corrected parts are highlighted in the manuscript and are mentioned according to the line numbers. Hope these improvements will make it acceptable for publication.

A point-by-point response to Reviewer #1 Comments:

The author has modified the experimental method and the results shown in the tables and figures, but the corresponding text has not been modified, such as L178, there is actually no COL in Table 1, and the LowAlc Kpn strain are not mentioned in the maintext.

I have nothing else to point out.

Response:

Thank you very much for your suggestions, which are very important to us.

i) We have added the data related to COL in the Table 1. Please refer to Table 1 for the changes.

ii) We are sorry about this error. In the discussion section, we added a description of low-alcohol-producing *K. pneumoniae* (LowAlc *Kpn*). Please refer to line 276-285 for the changes. We have already revised the sentence to “Meanwhile, we sought to investigate whether the glucose-induced polymyxins resistance mechanism was limited to HiAlc *Kpn*. The MIC and survival rate of low-alcohol-producing *K. pneumoniae* (LowAlc *Kpn*) under POL treatment in the presence 2% glucose were determined. When treated by glucose, LowAlc *Kpn* showed a higher-level resistance (MICs) to POL (Table 1). Consistent with the MIC test results, the survival rate of

LowAlc *Kpn* also increased significantly in the presence of glucose. And addition of cAMP or consumption of intracellular ATP by DCCD both could significantly inhibit the glucose-induced polymyxins resistance in LowAlc *Kpn* (Data was not shown). These results showed that glucose-induced polymyxins resistance was common in *K. pneumoniae*.”

May 31, 2023

Prof. Jing Yuan
Capital Institute of Pediatrics
Department of Bacteriology
No. 2 yabao road, Chaoyang District
Beijing, Beijing 100020
China

Re: Spectrum00031-23R2 (Glucose Induces Resistance to Polymyxins in High-alcohol-producing *Klebsiella pneumoniae* via Increasing Capsular Polysaccharide and Maintaining Intracellular ATP)

Dear Prof. Jing Yuan:

Your manuscript has been accepted, and I am forwarding it to the ASM Journals Department for publication. You will be notified when your proofs are ready to be viewed.

Sincerely,

Hui Wang
Editor, Microbiology Spectrum
